# The Histamine H_4_ Receptor Participates in the Anti-Neuropathic Effect of the Adenosine A_3_ Receptor Agonist IB-MECA: Role of CD4^+^ T Cells

**DOI:** 10.3390/biom11101447

**Published:** 2021-10-02

**Authors:** Laura Micheli, Mariaconcetta Durante, Elena Lucarini, Silvia Sgambellone, Laura Lucarini, Lorenzo Di Cesare Mannelli, Carla Ghelardini, Emanuela Masini

**Affiliations:** Department of Neuroscience, Psychology, Drug Research and Child Health (NEUROFARBA), Section of Pharmacology and Toxicology, University of Florence, Viale Gaetano Pieraccini 6, 50139 Florence, Italy; laura.micheli@unifi.it (L.M.); mariaconcetta.durante@unifi.it (M.D.); elena.lucarini@unifi.it (E.L.); silvia.sgambellone@unifi.it (S.S.); laura.lucarini@unifi.it (L.L.); carla.ghelardini@unifi.it (C.G.); emanuela.masini@unifi.it (E.M.)

**Keywords:** neuropathic pain, A_3_AR, H_4_R, allodynia, interleukin-10, CD4^+^ T cells, H_4_R^−/−^ mice, chronic constriction injury

## Abstract

A_3_ adenosine receptor (A_3_AR) agonists have emerged as potent relievers of neuropathic pain by a T cell-mediated production of IL-10. The H_4_ histamine receptor (H_4_R), also implicated in pain modulation, is expressed on T cells playing a preeminent role in its activation and release of IL-10. To improve the therapeutic opportunities, this study aimed to verify the hypothesis of a possible cross-talk between A_3_AR and H_4_R in the resolution of neuropathic pain. In the mouse model of Chronic Constriction Injury (CCI), the acute intraperitoneal co-administration of the A_3_AR agonist IB-MECA (0.5 mg/kg) and the H_4_R agonist VUF 8430 (10 mg/kg), were additive in counteracting mechano-allodynia increasing IL-10 plasma levels. In H_4_R^−/−^ mice, IB-MECA activity was reduced, lower pain relief and lower modulation of plasma IL-1β, TNF-α, IL-6 and IL-10 were shown. The complete anti-allodynia effect of IB-MECA in H_4_R^−/−^ mice was restored after intravenous administration of CD4^+^ T cells obtained from naïve wild type mice. In conclusion, a role of the histaminergic system in the mechanism of A_3_AR-mediated neuropathic pain relief was suggested highlighting the driving force evoked by CD4^+^ T cells throughout IL-10 up-regulation.

## 1. Introduction

The prevalence of neuropathic pain in the general population is estimated to lie between 6.9% and 10% [1]. Neuropathic pain refers to a broad range of clinical conditions that can be categorized anatomically (e.g., peripheral vs. central) and etiologically (e.g., degenerative, traumatic, infectious, metabolic, and toxic) [2]. Signs and symptoms associated with neuropathic pain include paresthesia, hyperalgesia, hypoalgesia, allodynia, ongoing pain (burning pain), paroxysmal pain (electrical shock-like pain), and abnormal temporal summation [3].

Current systemic and topical pharmacological treatments have substantial limitations in terms of the level of efficacy provided and/or the side effect profile. This means the management of neuropathic pain is unsatisfactory in preventing its development and progression [4]. This is why new pharmacological approaches are required.

Recently, the adenosine A_3_ receptor (A_3_AR) emerged as a novel target for neuropathic pain management. Preclinical studies demonstrated that A_3_AR agonists are effective in the prevention and treatment of neuropathies originated by different chemotherapeutic drugs like taxanes, platinum-complex and proteasome inhibitors [5,6] or by nerve trauma (e.g., the loose ligation of the sciatic nerve; Chronic Constriction Injury, CCI) [5,7]. A_3_ARs are expressed in both the peripheral [8] and central nervous systems [9], including glial cells, as well as in inflammatory and immune cells (i.e., macrophages and T cells). It is known that A_3_ARs are also located on the membrane of CD4^+^ T cells, a prominent source of IL-10 [10], and CD8^+^ T cells; their expression is increased under pathological settings correlated with the progression of the inflammatory response [11]. Moreover, the pivotal pharmacodynamic role of the CD4^+^ T-dependent IL-10 release in the pain-relieving effect of A_3_AR agonists was recently demonstrated [7].

The histamine H_4_ receptor (H_4_R), the last discovered histamine receptor subtype, also emerged as a promising target for pharmacological intervention in the development of new analgesics. Sanna and colleagues demonstrated that molecules acting on H_4_R are able to counteract neuropathic pain evoked by the spared nerve injury in mice, reducing both oxidative stress and pro-neuroinflammatory pathways [12,13]. Recent data showed the presence of H_4_Rs on neurons, highlighting its participation in nervous functions [12,14]. H_4_Rs are expressed in structures related to nociceptive transmission, such as dorsal root ganglia and spinal cord [12,14,15,16]. They have been also documented in several cell types of the immune system including T cells [17] where also A_3_ARs are expressed. H_4_Rs are implicated in the activation of CD4^+^ T cells [18] evoking the secretion release of different regulatory cytokines like IL-10 and IFN-γ [19]. On this basis, CD4^+^ T cells emerged as a possible crossroad between A_3_AR and H_4_R signaling suggesting synergistic or additive mechanisms in counteracting persistent pain.

The present work analyzed the relationship between the A_3_AR and H_4_R in modulating neuropathic pain induced by nerve trauma in the mouse model of loose ligation of the sciatic nerve. Wild type (WT) and H_4_R knockout mice were used to investigate the anti-hyperalgesic effects of the acute administration of the selective A_3_AR and H_4_R agonists, IB-MECA and VUF 8430, respectively, in comparison to their combination. The transplant of WT CD4^+^ T cells in H_4_R^−/−^ was used to highlight the role of the immune system in the H_4_R-dependent component of A_3_AR agonist pain relieving effect.

## 2. Materials and Methods

### 2.1. Animals

Male BALB/C WT and H_4_R^−/−^ mice (7 weeks old, 22–25 g starting weight) were used for the experiments. Histamine H_4_ receptor knockout mice were generated by Lexicon Genetics (Woodlands Park, TX, USA) and provided by Janssen Research and Development (LLC La Jolla, CA, USA) and back crossed to BALB/C background. Corresponding homozygous BALB/C WT controls were obtained from Envigo RMS S.r.l (Udine, Italy). Animals were housed in the Centro Stabulazione Animali da Laboratorio (University of Florence, Italy) and used at least 1 week after their arrival. Ten mice were housed per cage (size 26 × 41 cm); animals were fed a standard laboratory diet and tap water ad libitum and kept at 23 ± 1 °C with a 12 h light/dark cycle (light at 7 a.m.). The experimental protocol was carried out after approval by the Animal Care and Research Ethics Committee of the University of Florence, Italy, under license from the Italian Department of Health (No. 142/2017) and in compliance with international laws and policies (Directive, 2010/63/EU of the European parliament and of the council of 22 September 2010 on the protection of animals used for scientific purposes; Guide for the Care and Use of Laboratory Animals, US National Research Council, 2011).

Experiments involving animals have been reported according to ARRIVE guidelines [20]. All efforts were made to minimize animal suffering and to reduce the number of animals used.

### 2.2. Chronic Constriction Injury (CCI)-Induced Neuropathic Pain

CCI to the sciatic nerve of the left hind leg was performed under general anesthesia using the well-characterized Bennett and Xie model [21]. Briefly, animals (weighing 25–30 g at the time of surgery) were anesthetized with 2% isoflurane/O_2_ inhalation and maintained on 2% isoflurane/O_2_ for the duration of surgery. The left thigh was shaved and a small incision (1–1.5 cm in length) was made in the middle of the lateral aspect of the left thigh to expose the sciatic nerve. The nerve was loosely ligated around the entire diameter of the nerve at 3 distinct sites (spaced 1 mm apart) using silk sutures (6.0). The surgical site was closed with a single muscle suture and a skin clip. Pilot studies established that under our experimental conditions, the peak of mechano-allodynia develops by day 5–7 (D5–D7) following CCI. Test substances or their vehicles were given at peak of mechanical allodynia (D8–D9). A total of 70 animals underwent surgery for CCI.

### 2.3. Administration of Compounds

The selective A_3_AR agonist, IB-MECA (0.5–1 mg/kg; Tocris Bioscience, Milan, Italy), and the selective H_4_R agonist, VUF 8430 (10–30 mg/kg; Tocris Bioscience, Milan, Italy) were dissolved in sterile saline solution and intraperitoneally (i.p.) administered on day 8 after CCI surgery. Control animals received an equal volume of vehicle. Behavioral measurements were performed before and 30 min, 1 h, 2 h, 3 h and 5 h after compounds injection.

### 2.4. T Cells Isolation and Adoptive Transfer

Single-cell suspensions were obtained from spleens and lymph nodes of BALB/C WT mice by passing organs through 70 μm strainers, after which cells were washed with PBS plus 0.1% bovine serum albumin. T-cell population was purified by negative selection. Briefly, T cells were incubated with biotinylated antibodies against CD11b, CD11c, CD49b, B220, TER-119, CD4 and CD8a, all purchased from BioLegend (San Diego, CA, USA), and they were negatively selected by autoMACS sorting. After MACS purification, T cells were washed, counted and resuspended in PBS for intravenous (i.v.) injections (2 × 10^6^/200 μL/mouse). On day 7 after surgery, T cells or PBS were injected i.v. into the tail vein in a volume of 200 μL. An aliquot of the sorted population was assessed for the purity check analysis: cells were labelled with anti-CD3-FITC and the purity was determined by Flow Cytometry. The efficiency of transfer was confirmed by the restored presence of H_4_R in the spinal cord of H_4_R^−/−^ mice after the CD4^+^ T cells implantation (RT-PCR analysis; Appendix A).

### 2.5. Von Frey Test

The animals were placed in 20 × 20 cm plexiglass boxes equipped with a metallic mesh floor, 20 cm above the bench. A habituation of 15 min was allowed before the test. An electronic von Frey hair unit (Ugo Basile, Varese, Italy) was used: the withdrawal threshold was evaluated by applying force ranging from 0 to 5 g with a 0.2 g accuracy. Punctate stimulus was delivered to the mid-plantar area of each ipsilateral (injured side) hindpaw from below the mesh floor through a plastic tip and the withdrawal threshold was automatically displayed on the screen. Paw sensitivity threshold was defined as the minimum pressure required to elicit a robust and immediate withdrawal reflex of the ipsilateral hindpaw. Voluntary movements associated with locomotion were not taken as a withdrawal response. Stimuli were applied on each anterior paw with an interval of 5 s. The measure was repeated 5 times and the final value was obtained by averaging the 5 measures [22,23].

### 2.6. Cytokine Measurements

On day 8 after CCI surgery, WT and H_4_R^−/−^ mice were treated with IB-MECA, VUF 8430 or IB-MECA + VUF 8430 and 1 h post-dosing plasma samples were harvested to evaluate the levels of different cytokines. The quantitative determination of the interleukin-10 (IL-10), interleukin-6 (IL-6), interleukin-1β (IL-1β) and tumor necrosis factor (TNF)-α, was performed by a bead-based multiplex immunoassay, following the protocol provided by the manufacturer (EDM Millipore Corporation, Billerica, MA, USA). Briefly, neat plasma samples were added to antibody-conjugated beads directed against the cytokines listed above in a 96-well filter plate. After a 30 min incubation, the plate was washed, and biotinylated anti-cytokine antibody solution was added before overnight incubation. The plate was then washed, and streptavidin-conjugated PE was added.

After a final wash, each well was suspended with assay buffer and analyzed with the Bio-plex 200 system (Bio-Rad, Milan, Italy). Standard curves were derived from various concentrations of the different cytokine standards following the same protocol as the plasma samples.

### 2.7. Statistical Analysis

Data are expressed as mean ± SD for n animals. Behavioral data were analyzed by two-way repeated measures ANOVA followed by Dunnett’s test and Turkey comparisons. Calculation were made with Prism 6.1 statistical software (GraphPad Software Inc., San Diego, CA, USA). Electrophysiology averaged data are reported as mean ± SEM for n cells tested. Student’s paired *t*-test was used for statistical comparisons between data obtained from the same cell before and after treatment. Significant differences were defined as a value of *p* < 0.05.

## 3. Results and Discussion

In this study we showed a cross-talk between the molecular mechanisms underlying the effect of A_3_AR and H_4_R agonists in reducing neuropathic pain induced in mice by the loose ligation of the sciatic nerve. In particular, we demonstrated that submaximal doses of the selective A_3_AR agonist IB-MECA and the H_4_R agonist VUF 8430 showed additive effects in decreasing hypersensitivity. Further, IB-MECA efficacy partly depended from H_4_Rs with a mechanism involving the presence of CD4^+^ T cells and their ability to release the anti-inflammatory cytokine IL-10.

We reproduced the murine model of neuropathic pain induced by CCI, on day 8 after surgery the paw withdrawal threshold to non-noxious mechanical stimuli (von Frey test) was significantly decreased with respect to the same measure performed before the nerve damage (day 0). Intraperitoneal co-treatment with the selective A_3_AR (IB-MECA 0.5 mg/kg) and H_4_R (VUF 8430 10 mg/kg) agonists reversed mechanical allodynia in WT male mice with an effect that last up to 3 h after treatment (Figure 1). To note, the injections of the single agonists at the same dose used in the co-treatment were still active but did not reach the efficacy of the combination (Figure 1). The additive effect allowed to hypothesize a link between the mechanism of action triggered by the stimulation of both A_3_AR and H_4_R receptors. The dose of the A_3_AR agonist was chosen from previous studies to cause a near-to-maximal reversal of mechano-allodynia in this model [24], while the dose of the H_4_R agonist was selected on the basis of the results obtained after the acute administration of VUF 8430 and reported in the Appendix A.

To confirm the crosstalk between the histaminergic and adenosine system in the mechanism of action of IB-MECA, neuropathic WT and H_4_R^−/−^ mice were acutely treated with the A_3_AR agonist IB-MECA 1 mg/kg (the dose that exerts the full anti-hyperalgesic effect). In WT mice, IB-MECA fully counteracted CCI-induced mechanical allodynia 1 h after administration, the relief was significant up to 3 h (Figure 2). On the contrary, in H_4_R^−/−^ mice the compound was only partially effective (Figure 2) confirming a pivotal role of H_4_R in the anti-hypersensitivity action evoked by the A_3_AR agonist.

To investigate the possible interactions, we started to analyse one of the A_3_AR-mediated mechanisms: the attenuation of inflammatory cytokines, like TNF-α and IL-1β, and the increased formation of the anti-inflammatory IL-10 [25].

In our study, IB-MECA (1 mg/kg, 1 h after administration in CCI mice) significantly reduced IL-1β and TNF-α plasma levels in WT animals with respect to the vehicle-treated group (WT + vehicle), on the contrary in H_4_R^−/−^, IB-MECA did not modify the secretion of cytokines (Figure 3). Regarding IL-6 and IL-10, IB-MECA evoked a significantly higher increase in WT in comparison to H_4_R^−/−^ (Figure 4) reiterating the relevance of H_4_Rs in A_3_AR-mediated effects. Accordingly, IB-MECA and VUF 8430 were additive in increasing the plasma IL-10 concentration when administered at lower (per se ineffective) dose in WT (Figure 5).

On the other hand, the role of IL-10 in the A_3_AR-mediated pain relief overwhelmingly emerged since it was established that release of IL-10 from CD4^+^ T cells is required and sufficient for the A_3_AR agonists efficacy [7]. As a negative regulator, IL-10 is primarily produced by Th2 cells, activated B cells, monocytes, macrophages, and glial cells and regulates pleiotropic effects in inflammation and immunoregulation [26,27]. IL-10 is able to attenuate nociception in many animal models through the inhibition of pro-inflammatory cytokines and spinal glial cell activation [28]. Intrathecal IL-10 gene therapy has been shown to have a therapeutic effect in a rat model of neuropathic pain [29]. Moreover, IL-10 attenuates thermal hyperalgesia induced by chronic sciatic nerve constriction [30] and enhances morphine analgesia [31]. IL-10 is also a powerful neuroinhibitory cytokine; therapeutic manipulations aimed at increasing its presence in spinal cord (i.e., with plasmid DNA encoding IL-10) [32] or by indirectly increasing its production through the removal of peroxynitrite have been shown to block paclitaxel-induced neuropathic pain [33]. Andres-Hernando and colleagues demonstrated that IL-6 directly increases IL-10 production predominantly in the spleen and in CD4^+^ T cells participating to counteract the inflammatory response [34].

Focusing on CD4^+^ T cells, pivotal in the activity of A_3_ARs, it is interesting to note that also H_4_R is functionally expressed on these cells and implied in their activation [18] and the consequent production of different regulatory cytokines [17]. H_4_R activation was able to mitigate chronic inflammation increasing IL-10 and IFN-γ and inducing the recruitment of CD4^+^FoxP3^+^T cells [19]. So, hypothesizing CD4^+^ T cells as a potential site where the interaction between the two receptors takes place, WT mice were used to isolate CD4^+^ T cells that were intravenously administered in CCI H_4_R^−/−^ mice one day before the acute treatment with IB-MECA (1 mg/kg). The acute i.p. administration of IB-MECA reversed mechano-allodynia in H_4_R^−/−^ mice receiving CD4^+^ T cells in a comparable manner to that evoked by the same compound in WT mice (Figure 6) The baseline values showed that CD4^+^ T cells transfer did not influence the pain threshold of H_4_R^−/−^ mice.

## 4. Conclusions

In conclusion, we confirmed the crucial role of H_4_Rs on CD4^+^ T cells in modulating the antinociceptive responses of A_3_AR agonists. This effect seems to be regulated by the anti-inflammatory cytokine IL-10 enhancement. The selective stimulation of H_4_Rs together with low doses of A_3_AR agonists could have a clinical relevance for the treatment of neuropathic pain.

## Figures and Tables

**Figure 1 biomolecules-11-01447-f001:**
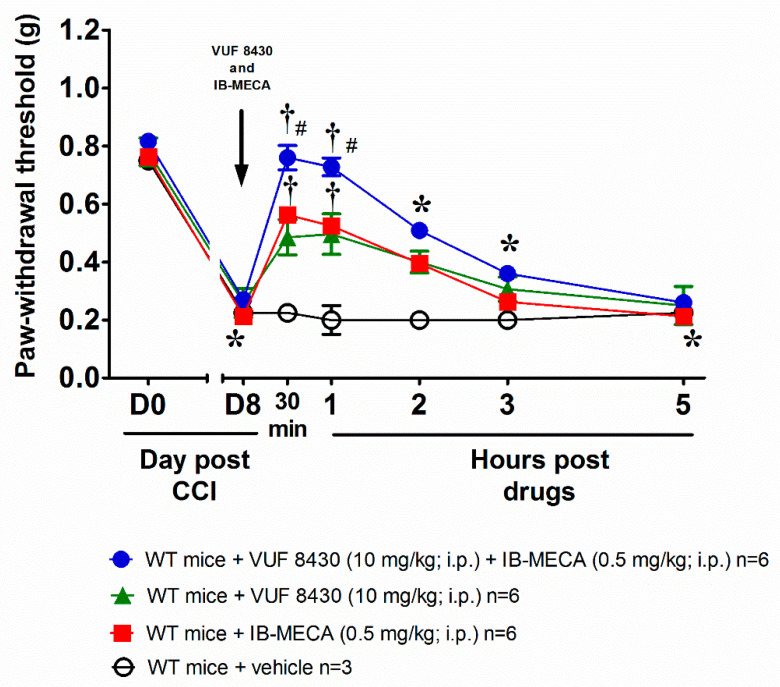
Anti-allodynic effect of A_3_ adenosine receptor (A_3_AR) and H_4_ histamine receptor (H_4_R) agonists on Chronic Constriction Injury (CCI)-induced neuropathic pain in wild type (WT) mice. Sciatic nerve ligation was performed 8 days before the acute injection of IB-MECA (0.5 mg/kg, i.p.), VUF 8430 (10 mg/kg, i.p.) or the combination of both (VUF 8430 10 mg/kg + IB-MECA 0.5 mg/kg). The response to a non-noxious mechanical stimulus was assessed by the von Frey test. Measurements were performed before and 30 min, 1 h, 2 h, 3 h and 5 h after compounds administration. Values reported in the graph are referred to the tests conducted on the ipsilateral paw. Data are mean ± SD for n mice per group; * *p* < 0.05 vs. Day 0 by two-way ANOVA with Dunnett’s test; † *p* < 0.001 vs. Day 8 by two-way ANOVA with Tukey’s comparisons; # *p* < 0.001 vs. WT + VUF 8430 (10 mg/kg) and WT + IB-MECA (0.5 mg/kg) by two-way ANOVA with Tukey’s comparisons.

**Figure 2 biomolecules-11-01447-f002:**
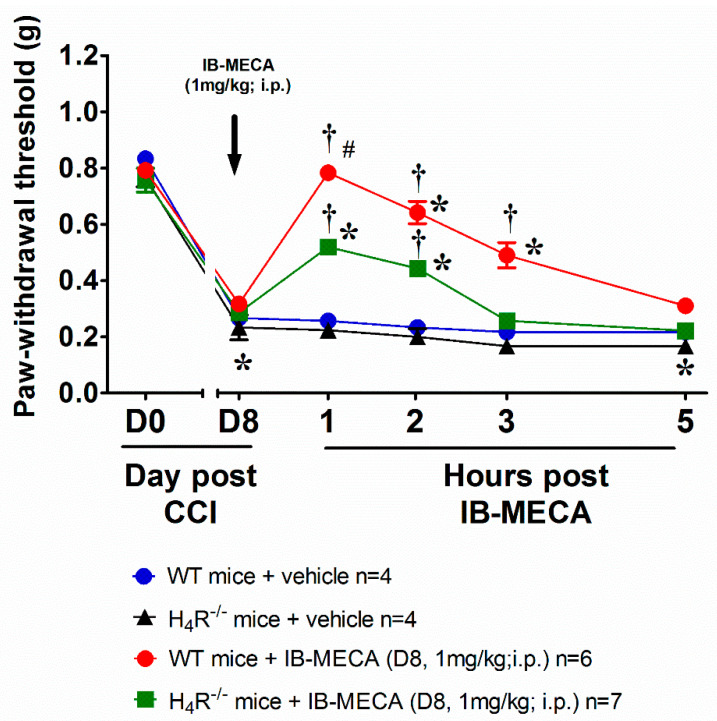
Anti-allodynic effect of A_3_AR agonist IB-MECA on CCI-induced neuropathic pain in WT and H_4_R^−/−^ mice. Sciatic nerve ligation was performed in WT and H_4_R^−/−^ mice 8 days before the acute injection of IB-MECA (1 mg/kg, i.p.). The response to a non-noxious mechanical stimulus was assessed by the Von Frey test. Measurements were performed before and 1 h, 2 h, 3 h and 5 h after IB-MECA administration. Values reported in the graph are referred to the tests conducted on the ipsilateral paw. Data are mean ± SD for n mice per group; * *p* < 0.05 vs. Day 0 by two-way ANOVA with Dunnett’s test; † *p* < 0.001 vs. Day 8 by two-way ANOVA with Turkey comparisons; # *p* < 0.001 vs. H_4_R^−/−^ mice + IB-MECA (1 mg/kg) by two-way ANOVA with Tukey’s comparisons.

**Figure 3 biomolecules-11-01447-f003:**
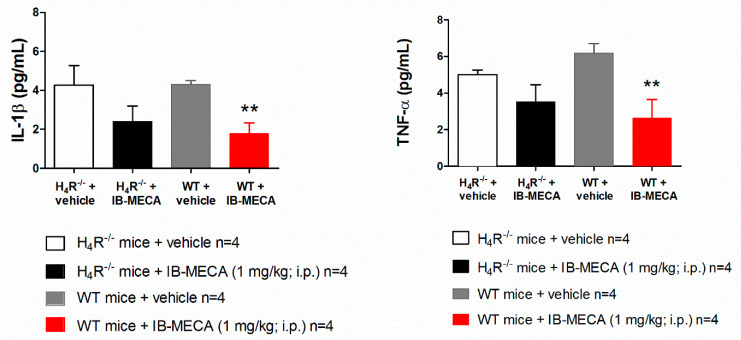
Effect of A_3_AR agonist IB-MECA on IL-1β and TNF-α plasma levels on CCI-induced neuropathic pain in WT and H_4_R^−/−^ mice. Sciatic nerve ligation was performed in WT and H_4_R^−/−^ mice 8 days before the acute injection of IB-MECA (1 mg/kg, i.p.). One hour after administration, blood was collected for dosing IL-1β and TNF-α plasma levels. Data are mean ± SD for 4 mice per group; ** *p* < 0.01 vs. WT mice + vehicle by one-way ANOVA with Dunnett’s test.

**Figure 4 biomolecules-11-01447-f004:**
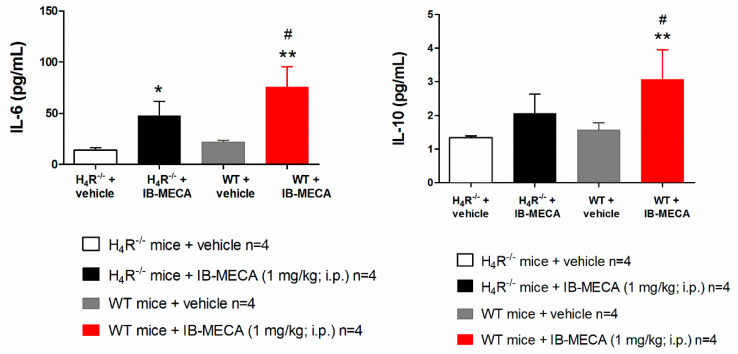
Effect of A_3_AR agonist IB-MECA on IL-6 and IL-10 plasma levels on CCI-induced neuropathic pain in WT and H_4_R^−/−^ mice. Sciatic nerve ligation was performed in WT and H_4_R^−/−^ mice 8 days before the acute injection of IB-MECA (1 mg/kg, i.p.). One hour after administration, blood was collected for measuring IL-6 and IL-10 plasma levels. Data are mean ± SD for 4 mice per group; * *p* < 0.05 and ** *p* < 0.01 vs. vehicle; # *p* < 0.05 vs. H_4_R^−/−^ mice + IB-MECA by one-way ANOVA with Dunnett’s test.

**Figure 5 biomolecules-11-01447-f005:**
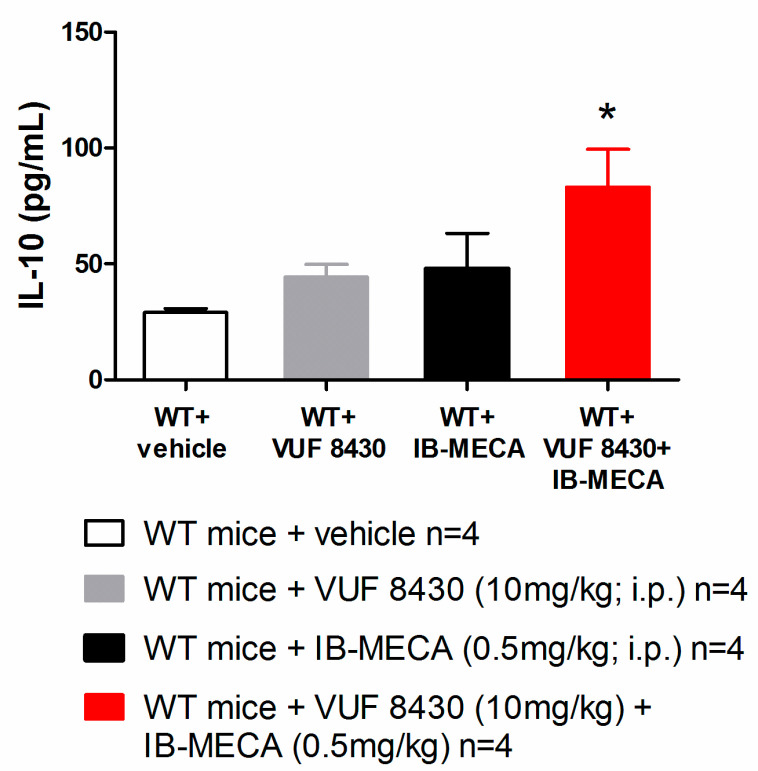
Effect of A_3_AR and H_4_R agonists on IL-10 plasma levels in WT mice underwent to CCI-induced neuropathic pain. Sciatic nerve ligation was performed 8 days before the acute injection of IB-MECA (0.5 mg/kg, i.p.), VUF 8430 (10 mg/kg, i.p.) or the combination of both (VUF 8430 0.5 mg/kg + IB-MECA 10 mg/kg). One hour after compounds administration, blood was collected for measuring the IL-10 plasma levels. Data are mean ± SD for 4 mice per group; * *p* < 0.05 vs. WT mice + VUF 8430 and WT + IB-MECA by one-way ANOVA with Tukey’s comparisons.

**Figure 6 biomolecules-11-01447-f006:**
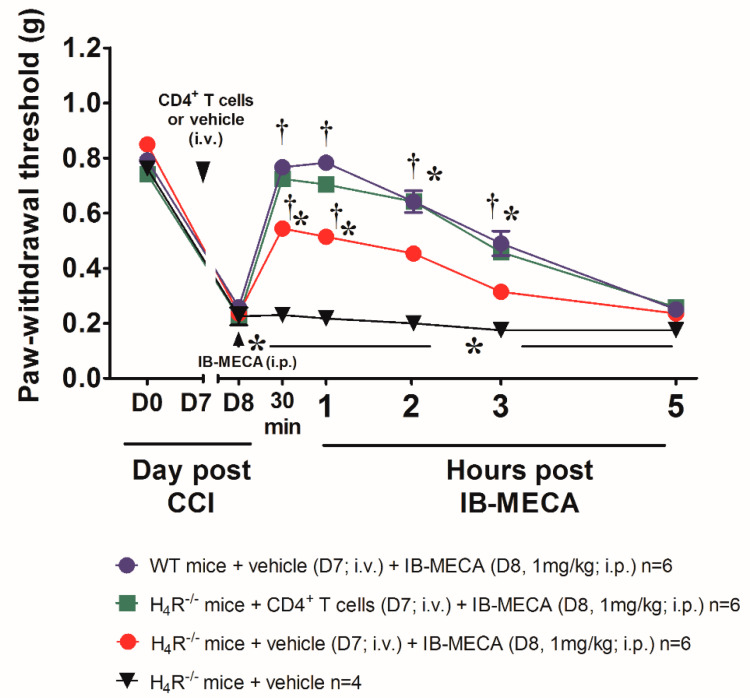
Effect of A_3_AR agonist IB-MECA after CD4^+^ T cells transfer in H_4_R^−/−^ mice underwent to CCI. Sciatic nerve ligation was performed in H_4_R^−/−^ mice 7 days before the intravenously administration of CD4^+^ T cells (2 × 10^6^/200 μL/mouse) obtained from naïve WT mice, 24 h the acute injection of IB-MECA (1 mg/kg, i.p.) was performed. The response to a non-noxious mechanical stimulus was assessed by the Von Frey test. Measurements were performed before and 30 min, 1 h, 2 h, 3 h and 5 h after compounds administration. Data are mean ± SD for n mice per group; * *p* < 0.05 vs. Day 0 by two-way ANOVA with Dunnett’s test; † *p* < 0.001 vs. Day 8 by two-way ANOVA with Tukey’s comparisons.

## Data Availability

The data presented in this study are available on request from the corresponding author.

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
