# Peer review of "The Histamine H4 Receptor Participates in the Anti-Neuropathic Effect of the Adenosine A3 Receptor Agonist IB-MECA: Role of CD4+ T Cells"

_biomolecules, 2021, doi:10.3390/biom11101447_

Round 1

Reviewer 1 Report

The authors Manneli et al. reported A3 adenosine receptor (A3AR) agonist, IBMECA, and H4 histamine receptor (H4R) agonist, VUF 8430, co-administered by intraperitoneal (i.p.) injection synergic reversed allodynic pain of CCI mice. In addition, CD4+ T cells played a modulation of A3AR activation-mediated pain relief. Generally, this report has some issues that required further address: 1. Line 18-20, This sentence should be rewritten for more clear description. 2. P2, 2.1. Animals, this section should be reformated in a more clear statement. 3. P3, 2.4. T cells isolation and adoptive transfer. It is good to know the immune response after T cell transplantation procedures. For example, the changes of cytokines in before and after T cells transfer. Moreover, the authors should provide the data or evidence of Flowcytometry to demonstrate the CD4+ T cells were been transplanted. 4. The descriptions, grouping, and group labels among each experimental group in this report are unclear. 5. P9, Figure 6. This experiment needs more control, such as WT mice + CD4+ T cell + IB-MECA, and H4R-/- mice + vehicle + IB-MECA. 6. There are many Greek letters showed in mistake symbols: lines 120, 139, 209, 212, 216, 218. 7. The format of references should check carefully before manuscript submitted: Ref. 5, 6, 12, 14, 15-17, 19, 20, 24, 29, 30, 33, 35-40, 42, 44, 45. 8. This manuscript was not prepared in concise and precise scientific descriptions. It is suggested to be reviewed by other colleagues before submission.

Reviewer 2 Report

Main Comments

Abstract

Line 17. Synergy can only be assessed through generation of an isobologram and this was not done. Hence, authors please replace ‘synergized’ by ‘were additive’

Materials and Methods

P2, subsection 2.1 entitled ‘Animals’, line 78. Why were the BALB/C wildtype control mice from a different breeding colony to the H4R-/- mice? This has the potential to introduce between-colony differences.

P2, subsection 2.2 entitled ‘Chronic constriction injury (CCI)-induced neuropathic pain’, line 95. How many mice underwent the CCI procedure?

P3, subsection 2.3 entitled ‘Administration of compounds’, line 107. How many mice per dose of each test compound were used?

Results and Discussion

P4, para 2, line 172. …..Replace ‘synergy’ by ‘additive effects’. The data in Figure 1 show that the effects of FUV8430 and IB-MECA appear to be additive rather than synergistic. As the authors did not construct an isobologram, they did not show synergy. Instead, they have shown ‘additivity’ between the effects of the adenosine A3 receptor agonist and the H4 receptor agonist in CCI-mice.

Figure 1 legend. Lines 181-182. The doses stated for the test compound combination have been inadvertently mixed up. ….combination of both (VUF8430 10 mg/kg + IB-MECA 0.5 mg/kg).

Line 185. State what ‘n’ is. This same comment applies to all of the other Figure legends.

Minor Comments

Title: ….participates in the…

Abstract

line 12. …also implicated in pain….

Line 18. ….anti-allodynia by increasing……IB-MECA was less active in….

Introduction

Para 2, line 39. ….progression [4]. This is why new….

P2, lines 62-63. …On this basis, CD4+ T cells….

Materials and Methods

P3, line 120. …200 μL.

P3, line 128. …Punctate…..area of each ipsilateral (injured side) hindpaw from…

P3, line 133. …on each ipsilateral hindpaw with….

P3, line 139. Interleukin 1β (IL-1β) and…

Results and Discussion

P4, line 174. …receptors. The dose of….

P6, line 212. ….reiterating the relevance of …

P6, line 213. …the plasma IL-10…

Figure 3 legend, line 218. …for measuring IL-1β and…

Figure 4 legend, line 223. … for measuring IL-6….

Figure 5 legend, line 230. … for measuring the…

Round 2

Reviewer 1 Report

I have no further comment.